# Did the COVID-19 Pandemic Restrict Access to Emergency Urological Services: Assessment of Reorganisation Effectiveness for Hospital Treatment

**DOI:** 10.3390/ijerph20043735

**Published:** 2023-02-20

**Authors:** Krystian Kaczmarek, Jakub Kalembkiewicz, Marta Jankowska, Karolina Kalembkiewicz, Jakub Narożnicki, Artur Lemiński, Marcin Słojewski

**Affiliations:** Department of Urology and Urological Oncology, Pomeranian Medical University, Powstańców Wielkopolskich 72, 70-111 Szczecin, Poland

**Keywords:** COVID-19, pandemic, urolithiasis, urinary stone disease, urological care

## Abstract

Urolithiasis derived renal colic is a common urological condition. If treated properly, the disease resolves without complications; if not treated, it causes infection and renal failure. The COVID-19 restrictions impacted hospitalised treatment of diseases. We analysed the impact of COVID-19 on renal colic treatment at a hospital in Poland. Clinical and demographic data of patients treated during the COVID-19 era were compared with those treated before this pandemic. During the COVID-19 restrictions, renal colic patient hospital admissions fell significantly. However, more patients presented with chronic renal colic symptoms and urinary tract infections. Nevertheless, the degree of hydronephrosis and the number and location of stones did not differ between the two groups. No marked changes were observed in the chosen treatment options. The observed decrease in emergency admissions of patients with acute renal colic with a simultaneous increase in the rate of infectious stones might indicate that some patients requiring urgent medical help did not report to the emergency department or came later than they would before the pandemic, reporting more serious symptoms. One plausible explanation for this may be that the reorganisation of the healthcare system restricted access to urological care. Moreover, some patients may have delayed their visit to the hospital due to the fear of contracting the SARS-CoV-2 coronavirus.

## 1. Introduction

Urolithiasis, characterized by the formation of stones in the urinary tract, is currently perceived as a public health concern worldwide [1]. Epidemiological studies indicate that the prevalence of this urinary stone disease varies widely between 0.1% to 18.5%, with developing countries experiencing prevalence rates of up to 25% of the population [2,3]. A wide range of disorders and pathogenetic mechanisms underlie the formation of stones in the urinary tract [4], whereas the symptomatology of urolithiasis depends mostly on the location of the stones (whether in the kidney, the ureter, or the urinary bladder) [3]. The acute passage of a stone from the kidney to the urinary bladder causes renal colic (an intense cramping pain), which is one of the most common acute urological conditions. Moreover, it is the leading cause of emergency room admissions. Renal colic is not life-threatening, and if properly treated, the disease runs its normal course without complications [5]. However, if treatment is delayed, urolithiasis may lead to severe infections, renal failure, and even death [6], whereas infected hydronephrosis might require urgent decompression to prevent further complications [7]. Therefore, adequate and timely treatment plays a key role in the management of acute episodes of urolithiasis. However, the world has been in an ongoing COVID-19 pandemic for more than 2 years that has influenced almost every aspect of life and led to the introduction of many restrictions, such as imposed lockdowns and social distancing. It has also affected healthcare systems in all countries worldwide [8]. The burden on the healthcare system caused by the treatment of patients infected with severe acute respiratory syndrome coronavirus (SARS-CoV-2) significantly contributes to delaying the treatment of patients with other clinical conditions, including those with urolithiasis [9]. Therefore, the objective of our study was to assess the impact of the COVID-19 pandemic on the presentation, course of disease, and treatment of acute episodes of urinary stone disease at the Department of Urology and Urological Oncology of Pomeranian Medical University in Szczecin, Poland. We hypothesized that the onset of the COVID-19 pandemic and the fear of getting a COVID-19 infection could possibly have led to a delay in patients with renal colic presenting to the emergency department and contributed to patients presenting with more severe clinical conditions at the time of hospital admission.

## 2. Materials and Methods

### 2.1. Study Methods

This single-centre, retrospective, observational and comparative study was exempted from further review by the Institutional Review Board (Bioethical Committee) of the Pomeranian Medical University, Szczecin, Poland, and was conducted in accordance with the regulations set forth by the Declaration of Helsinki. Patients included in this study routinely consented to participate in the study, specifically to allow the use of their anonymised treatment data for scientific purposes. We analysed consecutively admitted patients who were treated in the Department of Urology and Urological Oncology in Szczecin for an acute episode of urolithiasis during the first peak of SARS-CoV-2 infection in Poland (October–December 2020; Study group A). We compared them with patients admitted between October–December, 2019 (pre-COVID-19 era; Control group). In addition, to observe a changing pattern in the admission of patients with acute renal colic during the pandemic, we analysed the medical records of patients admitted consecutively during the second peak of COVID-19 cases in Poland (February–April 2021; Study group B). The selection of the analysed periods was based on the daily incidence of new SARS-CoV-2 cases in Poland. During the first peak of infection, the highest daily number of new cases was up to 30,000, whereas the highest recorded daily incidence during the second peak of the COVID-19 pandemic in Poland was 35,000 cases [10]. Throughout the pandemic, urological healthcare in Szczecin was significantly reorganised. The department that served as the focus of the present study was the only one to deal with emergency urological patients who did not suffer from COVID-19. Other urological departments in Szczecin were closed, and only one urological department was dedicated to hospitalising SARS-CoV-2-positive patients requiring emergency urological care.

For the present study, we only included patients who presented with symptoms of renal colic and required subsequent admission to the urological department. Indications for hospital admission included renal colic of a solitary kidney, bilateral renal colic, kidney injury, infected renal colic, intractable pain or nausea, and urinary extravasation. Before hospitalisation, each patient was examined by a urologist. Additionally, laboratory tests and urinary tract ultrasonography were performed to establish hydronephrosis. Finally, unenhanced computed tomography (CT) was performed to identify the location and size of the stones and to provide information regarding other potential aetiologies of pain. CT was also used to guide further management of the condition.

Data extracted from the medical records included age, gender, sex, body mass index (BMI), duration of symptoms, presence of fever, inflammatory markers, glomerular filtration rate (GFR), urinalysis, and urine and blood cultures. In addition, variables related to the presence of deposits on unenhanced CT, such as the size, location, and degree of hydronephrosis, were collected. Depending on the clinical presentation and images acquired, patients were qualified for further treatment: medical expulsive therapy (MET), urinary drainage (nephrostomy or double J stent implantation) with delayed definitive stone removal after the infection was cleared, or emergency removal of the deposit. Finally, all three analysed periods were compared to determine changing patterns in the clinical presentation of patients and management options chosen for acute episodes of urolithiasis after the reorganisation of emergency urological care due to the COVID-19 pandemic.

### 2.2. Statistical Analysis

Two independent reviewers checked the data for internal consistency. Descriptive statistics include mean and standard deviation (SD) for normally distributed data. Normality of distribution and homogeneity of variance were evaluated using the Shapiro–Wilk test and Levene’s test, respectively. Qualitative data are presented as numbers. A one-way analysis of variance (ANOVA) was performed to analyse the difference in the mean between the parametric variables. Non-parametric variables were compared using the non-parametric Kruskal–Wallis test. We considered a *p* value < 0.05 as statistically significant, and all *p* values were two-sided. All tests were performed using StatSoft statistical software, version 13.5 (StatSoft, Inc., Tulsa, OK, USA).

## 3. Results

The analysed cohort comprised 243 patients. The mean patient age was 54.99 (SD: 15.88) years. The female: male ratio was 105:138. The baseline characteristics of the patients included in the study are shown in Table 1. A significant difference was observed in the overall size of the sub-groups. The highest number of patients were admitted to the urological department during the pre-COVID period. The lowest number of patients were admitted at the second peak of the pandemic in the analyzed time period, which can now retrospectively be specified as the second highest peak of infections noted in Poland during entire the SARS-CoV-2 pandemic. The decrease in the admission of patients with renal colic between the pre-COVID-19 period and the first and second peaks of the pandemic was 35.90% and 56.41%, respectively. The daily admission rate fell from 1.3 patients per day in 2019 to 0.57 patients per day during the highest peak of the pandemic. Hospital length of stay (LoS) was analysed as a categorical variable (≤3 vs. >3 days). The median LoS for all analysed periods was 2 days. Notably, despite a downward trend in admission rate, no significant differences in hospital LoS were noted (*p* = 0.276). Thirteen patients required prolonged hospitalisation (>7 days) because of complicated renal colic. Five patients were hospitalised before the COVID-19 pandemic. Six and two patients were admitted during the first and second peak of SARS-CoV-2 infections, respectively.

Urological complaints reported at the time of admission differed between the analysed periods. During the pandemic, patients more often reported pain duration exceeding 7 days (*p* < 0.000; Table 1). The post hoc analysis indicated that there was no difference between the peaks of the pandemic. However, study groups A and B differed significantly from the control group. Moreover, the presence of fever at admission was significantly higher in patients during the COVID-19 pandemic compared to the control group (*p* = 0.003; Table 1). The post-hoc tests indicated that the difference was only observed between study group B and the control group. It is highly important to underline that every patient was tested for COVID-19 at admission to our Department. None of the admitted patients that were included in the study was infected. Therefore, the fever at admission was not caused by the SARS-CoV-2 infection. It was also noted that during the pandemic period, patients were statistically more likely to present elevated inflammatory parameters such as C-reactive protein (CRP; *p* < 0.001), procalcitonin (PCT; *p* < 0.001), and leucocytosis (*p* < 0.001, Table 2). A post hoc analysis highlighted that CRP and PCT levels were significantly different between the study groups (A and B) and the control group. Leucocytosis was higher in subgroup B than that in the control group. Also, we observed changes in the parameters of urinalysis. During the pandemic, more patients presented with leukocyturia (*p* = 0.019) and nitrite-positive urine at the time of admission (*p* = 0.01; Table 2).

There was no significant difference in the degree of hydronephrosis (*p* = 0.250), the number (*p* = 0.557), or the location of stones (*p* = 0.575) between the analysed groups. However, there was a trend that patients hospitalised during the COVID-19 pandemic period (study groups A and B) had more frequent decompression of the urinary tract by the implementation of nephrostomies and double-J catheters compared to patients from the pre-pandemic period (control group).

## 4. Discussion

The first Polish COVID-19 patient was diagnosed on 4 March 2020 [10]. To date, our country has faced five significant waves of this pandemic, especially during the first year after the outbreak, which caused a major reorganisation of the Polish healthcare system. Despite considerable efforts to maintain continuity in specific treatments, particularly for oncological and emergency conditions, many diseases remained untreated [11]. Several studies have attempted to analyse the course of urolithiasis during the COVID-19 pandemic in different countries around the world. To date, only a few similar studies concerning the impact of the pandemic on urological services have been conducted in Poland [12,13]. The largest study was a multicentre study that tried to assess the changing patterns of urologic emergency visits and admissions during the COVID-19 pandemic. However, in the aforementioned study, a pooled analysis of all urological conditions was performed without a dedicated urolithiasis assessment. Other studies performed by Rajwan et al., did not provide information regarding the reorganisation of healthcare systems in particular cities [13]. Therefore, in this study, we analysed the changing pattern in the clinical presentation of patients with acute episodes of renal colic. In our city, there were unique reorganisations of the healthcare system when only one department covered all emergency urological patients who did not suffer from COVID-19. Thus, we can observe how this model of reorganisation worked in Szczecin, a city with more than 400,000 inhabitants.

Recent studies have reported that the reduced provision of medical services and limited access to surgical care affect the course of urinary stone disease. There has been a significant reduction in the number of emergency visits and admissions for urolithiasis [14,15,16]. Polish researchers also noticed a reduction in emergency room admissions nationwide during the national lockdown [13]. Thus, a 35.90% decrease from 117 in 2019 to 75 admissions in 2020 and a 56.41% decrease to only 51 patients with renal colic admitted to our department are in line with previous reports. The lowest number of patients with symptomatic urolithiasis occurred during the period with the highest peak of COVID-19 infection. A significant reduction in the number of patients presenting with renal colic in the pandemic era could be the result of fear of exposure to SARS-CoV-2 and many COVID-19-related restrictions in our country. Available publications have already analysed the factors affecting the reduction in urgent admissions and indicated that the main concern of patients was the perception of hospitals as reservoirs of COVID-19 and the fear of getting infected [17].

It is worth noting that the duration of symptoms before the visit to the emergency department in our patients was significantly longer, and more patients reported fever at admission. Moreover, a higher proportion of patients had elevated inflammatory parameters. Taken together, our findings suggest a more severe course of urolithiasis in the COVID-19 era. Surprisingly, Byrne et al., not only observed fewer patients suffering from fever at admission but also found no significant differences in mean levels of CRP, white cell count, positive microbiological cultures (urine or blood), and creatinine [18]. Conversely, in another worldwide study, higher serum creatinine levels, increased grade 3 and 4 hydronephrosis, and higher incidence of leucocytosis and complications in comparison to the pre-COVID-19 period were observed [10]. Due to the overloaded healthcare system, we recommended that patients contact emergency medical services via remote means of communication, such as phone calls or virtual sessions, and appear at emergency departments only if they present with uncontrolled pain or fever. Therefore, we speculate that these factors may have significantly contributed to the delay in reporting to urological services, and that this may offer a plausible explanation for the more severe course of urolithiasis in our study groups. However, considering the multicentre analysis of urological emergency visits and admissions, other Polish researchers did not find any significant differences in laboratory parameters [13]. The disparate results of these reports and our present study suggest that the symptoms and severity of urolithiasis during the pandemic era might differ among various urology departments. This in turn may be explained by the varying numbers of COVID-19 cases in different regions of Poland and differences in the reorganisation of the healthcare system. Hence, it is crucial to examine the situation in every medical centre individually to assess the potential needs of patients.

Our current data also show a lack of difference in maximum stone size between the pre-pandemic and pandemic periods. Moreover, we did not observe any differences in the number of stones and location. This might be justified by the fact that even small-sized deposits can produce exaggerated symptoms of obstruction and cause patients to report to the hospital. Similar results have been reported in other studies [18,19,20]. However, Jiang et al., reported that the mean stone size among patients visiting the emergency room in the pre-COVID and COVID periods increased from 5.1 to 10.5 mm. The authors explained this to be due to the significant impact of the pandemic on the preferences of patients for urolithiasis therapy. During the pandemic, patients with symptomatic stones expressed reluctance to undergo procedural intervention. Patients opted instead for at-home conservative treatment and reserved emergency room visits for larger stones, potentially causing self-harm [21].

In our study, we only analyzed acute admissions related to urolithiasis. We did not consider data on acute admissions related to other acute urological conditions, and we do not know whether similar data have been collected on other types of acute surgical admissions by the surgical departments of our hospital. Nevertheless, some research has already been carried out in Poland on this topic. Interesting results can be seen in the work of Pawelczyk et al., where two groups of pediatric patients, admitted before and during the pandemic for acute appendicitis were compared. The difference in the time between the onset of symptoms and presentation to the hospital draws ones attention. Significantly more patients were admitted to the surgery department after a longer time from the onset of symptoms during the pandemic. Additionally, there was a decrease in phlegmonous and gangrenous cases in favor of an increase in complicated cases of diffuse peritonitis due to perforation of the necrotic appendix [22]. Therefore, patients in the aforementioned study had a more severe course of the disease, which is similar to our observations considering urolithiasis.

Despite these important findings, our study has several limitations. First, our study was restricted by constraints inherent to the retrospective nature of the data analysis. Therefore, we were unable to control all confounding factors that may have influenced the laboratory results in our analyses. Additionally, it should be noted that because our study was based on in-patient data, we were unable to estimate the exact number of patients with renal colic who required only online consultation or outpatient care. Therefore, the rate of acute episodes of renal colic could be underestimated because many patients were treated as out-patients. Despite these limitations, we believe that our results will be helpful in understanding the changing patterns of the clinical presentation of patients with acute conditions related to urolithiasis during the COVID-19 pandemic. Consequently, our findings will be useful in planning a better reorganisation of healthcare systems in the future, potentially resulting in an improvement in patient safety as well as reduced healthcare expenditures.

## 5. Conclusions

Our data show that the number of patients admitted to the hospital for renal colic decreased as the number of SARS-CoV-2 infections increased. Most patients presented with prolonged renal colic symptoms before visiting the emergency department and were more likely to have a urinary tract infection. A plausible explanation for this phenomenon is that the reorganisation of the healthcare system may have restricted the access of patients to urological care. Moreover, patients might have come to the hospital later due to their fear of contracting the SARS-CoV-2 coronavirus.

## Figures and Tables

**Table 1 ijerph-20-03735-t001:** Baseline patients’ characteristics.

Parameters	Control Group	Study Group A	Study Group B	*p* Value
Time period	Oct.–Dec. 2019	Oct.–Dec. 2020	Feb.–Apr. 2021	
Totals, No.	117	75	51	
Age, years				0.691
Mean	54.889	54.08	56.549	
SD	15.982	15.359	16.574	
Gender				0.168
Female	51	27	27	
Male	66	48	24	
BMI (kg/m^2^)				0.242
<30	67	48	36	
≥30	50	27	15	
Fever				0.003
no	113	66	41	
yes	4	9	10	
Duration of symptoms (days)				0.000
≤7	94	34	11	
>7	23	41	40	
Length of stay (days)				0.276
≤3	82	59	34	
>3	35	16	17	

BMI: Body Mass Index; SD: standard deviation.

**Table 2 ijerph-20-03735-t002:** Crucial laboratory parameters and imaging findings in patients admitted due to renal colic.

Parameters	Control Group	Study Group A	Study Group B	*p* Value
Time period	Oct.–Dec. 2019	Oct.–Dec. 2020	Feb.–Apr. 2021	
eGFR				0.351
≥90	20	17	15	
60–89	51	28	18	
30–59	38	24	18	
15–29	6	4	0	
<15	2	2	0	
C-reactive protein				0.000
Median	8.17	39.58	60.00	
Range	0.28–422.13	0.34–297.43	7.88–298.40	
Procalcitonin				0.001
Median	0.09	0.11	0.50	
Range	0.02–19.00	0.02–20.40	0.02–96.80	
White blood cell count				0.000
Median	8.74	9.37	13.20	
Range	3.80–22.89	4.53–29.92	4.50–47.00	
Erythrocyturia				0.586
no	56	34	20	
yes	61	41	31	
Leukocyturia				0.019
no	70	33	20	
yes	47	42	31	
Nitrite-positive urine				0.011
no	107	66	38	
yes	10	9	13	
Urinary culture				0.205
negative	102	58	42	
positive	15	17	9	
Blood culture				0.454
negative	107	72	48	
positive	10	3	3	
Grade of hydronephrosis				0.250
no	10	11	4	
renal pelvis	88	38	36	
renal calyces	19	26	11	
Stone size (mm)				0.094
<5	47	46	25	
5–10	40	8	21	
10–20	20	13	4	
>20	10	8	1	
Number of stones				0.557
single	103	63	46	
multiple	14	12	5	
Stone location				0.575
renal pelvis	5	1	3	
upper ureter	35	27	16	
middle ureter	27	24	11	
distal ureter	50	23	21	
Type of intervention				0.237
MET	17	13	1	
Urinary drainage	6	10	11	
Stone removal	94	52	39	

eGFR: estimated glomerular filtration rate; MET: medical expulsive therapy; SD: standard deviation.

## Data Availability

Source data available at: https://osf.io/4a93j, accessed on 14 January 2023.

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
