# Peer review of "Did the COVID-19 Pandemic Restrict Access to Emergency Urological Services: Assessment of Reorganisation Effectiveness for Hospital Treatment"

_ijerph, 2023, doi:10.3390/ijerph20043735_

Round 1

Reviewer 1 Report

Maybe, it would be useful to compare the delay at admittance with other acute surgical diseases during the COVID-19 pandemic conditions, if such data are available. Is it different?

Reviewer 2 Report

Title: Did the COVID-19 pandemic restrict access to emergency uro-2 logical services: Assessment of reorganisation effectiveness for 3 hospital treatment

The study has a sound methodology.

The analysis has been described well.

The discussion is clear and balanced.

I have a few minor comments:

Line 120          Which peak are the authors referring to? The first or second COVID peak included in the study.

Line 141          The authors have mentioned that 'the presence of fever at admission was significantly higher in patients during the COVID-19 pandemic compared to the control group.' Were the patients infected with SARS-CoV-2 during the time of admission? It would be noteworthy to add the information.

Reviewer 3 Report

After reviewing the document, we see that attention is focused on urological care that was diminished by the effect of the pandemic, which largely delayed the care of patients, conditioning them more severe conditions, as well as these themselves due to fear of getting sick. They did not come, worsening their symptoms

The importance lies in the fact that this same affectation, as an example, can be extrapolated to other medical care that surely were equally diminished, the most serious being chronic and degenerative patients, however non-chronic pathologies can likewise contribute to the aggravation or appearance of diseases of this type

I consider that, despite being a very specific point -urological-, it reveals the ravages of the pandemic at the hospital level, which with the hospital conversion reduced many clinical and surgical programs affecting the population

I consider it can be published
